# A Novel Fabrication of Heterogeneous Saponified Poly(Vinyl Alcohol)/Pullulan Blend Film for Improved Wound Healing Application

**DOI:** 10.3390/ijms25021026

**Published:** 2024-01-14

**Authors:** Sabina Yeasmin, Jae Hoon Jung, Jungeon Lee, Tae Young Kim, Seong Baek Yang, Dong-Jun Kwon, Myoung Ok Kim, Jeong Hyun Yeum

**Affiliations:** 1Department of Biofibers and Biomaterials Science, Kyungpook National University, Daegu 41566, Republic of Korea; yeasminsabina44@yahoo.com (S.Y.); jjdwognsz@naver.com (J.H.J.); dlwjddjs2@gmail.com (J.L.); xodud10301@naver.com (T.Y.K.); 2Research Institute for Green Energy Convergence Technology, Gyeongsang National University, Jinju 52828, Republic of Korea; sbyang@gnu.ac.kr (S.B.Y.); rorrir@empas.com (D.-J.K.); 3Department of Materials Science and Convergence Technology, Gyeongsang National University, Jinju 52828, Republic of Korea; 4Department of Animal Science and Biotechnology, Kyungpook National University, Sangju 37224, Republic of Korea; ok4325@knu.ac.kr

**Keywords:** poly(vinyl acetate), poly(vinyl alcohol), blend film, casting process, heterogeneous saponification, hydrophilic film, degree of saponification

## Abstract

In this study, a novel film of poly(vinyl alcohol) (PVA)/pullulan (PULL) with improved surface characteristics was prepared from poly(vinyl acetate) (PVAc)/PULL blend films with various mass ratios after the saponification treatment in a heterogeneous medium. According to proton nuclear magnetic resonance (^1^H-NMR), Fourier transform infrared, and X-ray diffraction results, it was established that the successful fabrication of saponified PVA/PULL (100/0, 90/10, and 80/20) films could be obtained from PVAc/PULL (100/0, 90/10, and 80/20) films, respectively, after 72 h saponification at 50 °C. The degree of saponification calculated from ^1^H-NMR analysis results showed that fully saponified PVA was obtained from all studied films. Improved hydrophilic characteristics of the saponified films were revealed by a water contact angle test. Moreover, the saponified films showed improved mechanical behavior, and the micrographs of saponified films showed higher surface roughness than the unsaponified films. This kind of saponified film can be widely used for biomedical applications. Moreover, the reported saponified film dressing extended the lifespan of dressing as determined by its self-healing capacity and considerably advanced in vivo wound-healing development, which was attributed to its multifunctional characteristics, meaning that saponified film dressings are promising candidates for full-thickness skin wound healing.

## 1. Introduction

An extracellular microbial polysaccharide called pullulan (PULL) is produced by various strains of *Aureobasidium* [1,2]. It has attracted attention in biomaterials science due to its remarkable characteristics [3,4]. Similar to the common benefits of natural polysaccharides, PULL possesses good water solubility, high water-absorbing ability, and properties that produce strong, resilient films and fibers due to its robust chemical structure [4]. The films and fibers produced from PULL are tasteless, colorless, odorless, clear, and bendable [5]. Moreover, the wound-healing characteristics of PULL make it an appropriate candidate for wound dressings in burn and wound care [6]. Wound-dressing material should show biodegradable, biocompatible, mucoadhesive, hemostatic, and bactericidal characteristics [7]. Among various wound-dressing materials, pullulan has numerous advantages; for example, along with the abovementioned characteristics, the hydrophilic groups of PULL create a three-dimensional crosslinked network. Research has demonstrated that PULL functions as an immunomodulatory substance that controls wound inflammation [8,9]. However, the low mechanical property of PULL necessitates the need for blending with other polymers [10]. Its use in a pure form for films has been limited by the films’ poor mechanical behavior and expensiveness [10]. One useful method to minimize the cost and advance the properties of PULL-based films is to blend PULL with comparatively cheaper and compatible polymers with improved mechanical behavior. Blending is a usual and potentially versatile method to obtain new films with improved characteristics [11]. Sometimes, a polymer’s hydrophobic nature restricts its ability to interact with living cells, which prevents the polymer from adhering to the compromised area and causing insufficient cell proliferation. In order to maintain the proper moisture levels in the injured area and thereby speed up the healing process, the hydrophilicity of the polymer is increased by mixing it with hydrophilic polymers [12]. In this study, blending PULL with poly(vinyl alcohol) (PVA) can improve surface properties and hydrophilic characteristics that support the absorption of excess wound exudates and retention of moisture at wound sites [13].

Homopolymer poly(vinyl acetate) (PVAc) is synthesized from vinyl acetate (VAc) monomer using a free radical polymerization technique. Synthetic polymer PVAc is applied to identify gas, moisture, and humidity. Moreover, PVAc films are applied for oral drug delivery and as emission sensors for transportation [14]. In addition, PVA is considered a good candidate to blend with natural polymer due to its good mechanical characteristics, exceptional chemical resistance, biodegradable properties, easy preparation, and film-forming capability [15,16].

PVA, a linear semicrystalline polymer produced by saponifying poly(vinyl ester), is extensively utilized in the textile, film, and membrane industries as well as in drug delivery systems [17]. Generally, if there are no inorganic material substituents and no surface treatment is applied, the PVA film’s surface is smooth. To modify the properties of PVA, the heterogeneous saponification method was used by several researchers to convert PVAc to PVA, and the prepared PVA showed improved properties [18]. According to previous studies, saponified PVA shows unusual surface characteristics compared with the standard PVA, which is considered beneficial for various applications [19,20]. Besides, biological organisms can readily degrade polyvinyl alcohol (PVA), which is essentially produced by hydrolyzing polyvinyl acetate [21]. Additionally, the saponification method can be used to prepare hydrophobic–hydrophilic PVA blend films [22], which are typically challenging to fabricate due to variations in cosolvent solubility, poor compatibility between the hydrophobic matrix and hydrophilic polymer, and the possibility of phase separation during the film production process [23,24]. PVA and PULL blending generally exhibit immiscibility [13], which can be resolved by applying the suggested saponification method.

Suitable and fine wound care is critical, particularly in the presence of pathology. The increased occurrence of obesity and diabetes further highlights the advantages and need for better wound care. According to a recent study, wound healing is assisted by a moist dressing [25]. Moist wound healing is a therapeutic process with exudates as the humectants to defend and offer moist surroundings for the wound. Usually, hydrophilic materials, such as cellulose, are used in clinical wound care [2].

In this study, we developed saponified PVA/PULL films with various blend ratios and compared their characteristics. First, PVAc/PULL film was prepared using the solution casting method; subsequently, the prepared film was converted to PVA/PULL film using the saponification method, which was verified by a proton nuclear magnetic resonance (^1^H-NMR), Fourier transform infrared (FT-IR), and X-ray diffraction (XRD) analyses. The prepared heterogeneous saponified PVA/PULL film showed some unique characteristics, such as an uneven surface with high surface area [20], improved hydrophilic properties, and moisture stability due to the repacking of polymer chains [26], which support the absorption of excess wound exudates and retention of moisture at wound sites. Besides, it protects wounds from outside harmful microorganisms.

## 2. Results and Discussion

### 2.1. OM

The morphologies of the upper surfaces of the saponified and unsaponified films were inspected by OM, as observed from the micrographs shown in Figure 1. Figure 1a–c depict the micrograms of unsaponified PVAc, PVAc/PULL 90/10 blend, and PVAc/PULL 80/20 films, respectively, whereas Figure 1d–f depict the micrograms of saponified PVAc, PVAc/PULL 90/10 blend, and PVAc/PULL 80/20 films, respectively. The unsaponified PVAc and PVAc/PULL 90/10 blend films show smoother surfaces than the unsaponified PVAc/PULL 80/20 blend film; however, no phase separation can be seen within the resolution of the method. The micrographs of saponified films displayed higher surface roughness than those of the unsaponified films, which might be due to the dissolution of film skin in the alkaline solution and the increase in film density during the drying of the film after saponification [20].

### 2.2. SEM

Figure 2 shows SEM images taken at the same magnification that specify the unsaponified and saponified PVAc/PULL blend films with different blend ratios of 100/10, 90/10, and 80/20. As anticipated, we found that following saponification, the surface roughness of all kinds of films increases (Figure 2d–f). Thus, following saponification, the surface area also increased. PVA/PULL skins that dissolve in the aqueous saponification solution and an increase in film density during drying could be the causes of the change in the film surface following saponification [20]. As the PULL mass ratio in the PVAc/PULL blend film increased, we also observed that the film surface became rougher (Figure 2a–f), perhaps as a result of the increased PULL content interfering with the PVAc’s continuity.

### 2.3. AFM

To study the effect of saponification on the evolution of film morphology, AFM was performed to investigate the surface morphology of the blend films (Figure 3). Surface geometry can be examined using many methods, but only AFM enables the sampling of real surface heights, which use forces that happen during direct contact between the exterior and scanning tip. Figure 3 reveals the AFM images of the samples under study that are 150–200 µm thick and scanned over areas of 5 × 5 µm^2^. Reassuringly, the film surface root-mean-square values for pure PVAc increased from 6.534 to 16.003 nm after the saponification treatment, whereas those for PVAc/PULL 90/10 changed from 1.982 to 29.79 nm, indicating the formation of a more nonuniform film surface; a similar tendency was observed for the PVAc/PULL 80/20 film.

### 2.4. FT-IR Spectroscopy

To investigate the structural change in the PVAc/PULL film after heterogeneous saponification, FT-IR analysis was performed. FT-IR data for saponified and unsaponified PVAc/PULL films with various mass ratios of 100/0, 90/10, and 80/20 are shown in Figure 4. According to Figure 4a, asymmetric and symmetric methyl stretch vibration bands of pure PVAc were observed in the range of 2800–2900 cm^−1^. Meanwhile, the saponified PVAc film showed a –OH stretch in the range of 3000–3600 cm^−1^ that originated from both intermolecular and intramolecular hydrogen bonds, which meant the acetyl group (–OCOCH_3_) of PVAc film turned to a hydroxyl group (–OH) due to the reaction with alkaline solution during heterogeneous saponification, which revealed that PVA was obtained from PVAc. A similar result was also found in our previous studies [20,27]. Figure 4b,c also show a similar trend regarding the saponified films, but the peak intensity varied depending on the PVAc/PULL ratio, which might be due to the change in material property. However, when FT-IR spectra of the unsaponified films were considered, Figure 4c shows a very precise broad peak in the range of 3000–3600 cm^−1^, as PULL also contains the –OH group [28]; it was also revealed that PVA was obtained from PVAc in the presence of PULL.

### 2.5. XRD

The XRD patterns of unsaponified and saponified PVAc/PULL (100/0, 90/10, and 80/20) films are shown in Figure 5. According to Figure 5, two broad peaks of unsaponified PVAc observed at 13.5° and 22.5° revealed the amorphous characteristics of the pure PVAc film [20]. Meanwhile, the saponified PVAc films showed a sharp crystalline peak at 2θ = 19.8° [19], which is like the characteristic peak of standard PVA, i.e., PVAc was successfully converted to PVA. Similar behavior was observed for the saponified PVAc/PULL (90/10, 80/20) blend films (Figure 5b,c); however, the PVA peak intensity decreased depending on the PVAc content of the blend films. Moreover, various crystalline peaks that appeared for the saponified blend films might be due to the crystallization of PULL during saponification, and their variations with different blend ratios are due to the effect of PULL concentration on PULL crystallization. Similar behavior was observed in another research work, where the crystallization of sucrose occurred when stored in humid conditions [29].

In addition, the unsaponified PVAc/PULL (100/0, 90/10, and 80/20) blend films’ peak position and intensity changed depending on the blend ratio. These results indicate the successful fabrication of PVAc/PULL blend films with varying mass ratios (90/10 and 80/20).

### 2.6. ^1^H-NMR Spectroscopy

The ^1^H-NMR spectra for the unsaponified and saponified pure PVAc film and PVAc/PULL blend films with various mass ratios of 90/10 and 80/20 are shown in Figure 6. The degree of saponification (DS) was calculated according to reference [16] using the following equation:Degree of saponification (%) = 100 − 100{(Peak area of CH_3_/3)/(Peak area of CH_2_/2)}(1)

The methyl peak was absent, and methylene peaks appeared after saponification for all films, i.e., the successful conversion of PVAc to PVA occurred. From the DS value, fully saponified PVA was obtained from all studied films. Notably, the presence of PULL did not affect the saponification; it even increased DS slightly, and there was no difference between the blend films (PVAc/PULL 90/10 and PVAc/PULL 80/20) regarding DS. A DS of 99.9% was obtained for PVAc/PULL 90/10 and 80/20 films, whereas PVAc showed 99.5% DS. Finally, it is established that fully saponified PVA and PVA/PULL (90/10 and 80/20) could be obtained from PVAc and PVAc/PULL (90/10 and 80/20), respectively, after 72 h of saponification at 50 °C.

### 2.7. Contact Angle Analysis

Finally, the wettability properties of the saponified and unsaponified films were evaluated using contact angle measurement. As shown in Table 1, a decrease in the contact angle was seen in all saponified films compared with unsaponified films. Besides, a clear hydrophilic improvement with the increase in PULL content in blend films was observed. Another research group had observed a similar behavior [30]; as they added more PULL inside the studied blend nanofiber film, the contact angle declined. Moreover, in this study, the decrease in contact angle of the saponified film was due to the conversion of PVAc to PVA, as PVA is a well-known hydrophilic material, whereas PVAc is hydrophobic. So, it is confirmed that saponified films become more hydrophilic after saponification. In addition, the presence of PULL added extra hydrophilicity to the saponified PVAc/PULL film compared with the unsaponified PVAc/PULL film.

### 2.8. TGA

TGA was applied to investigate the dehydration and decomposition of the studied films. Figure 7 and Figure 8 show the TGA and derivative thermogravimetric (DTG) curves of saponified and unsaponified PVAc/PULL (100/0, 90/10, and 80/20) films. According to the figures, all films showed three thermal decomposition stages. The first weight loss occurred in the range of 70–200 °C, and about 7–8% weight loss was due to surface-absorbed water and intermolecular-bound water, which evaporated due to heat [31,32]. The second weight loss for saponified PVAc was about 68%, ending at about 336.24 °C, and the highest weight loss rate that occurred at 220.08 °C might have been due to the break and disintegration of volatile products and dihydroxylation of the saponified PVAc film [33]. The second-stage weight loss for the saponified PVAc/PULL 90/10 blend film was about 61%, beginning at 215.04 °C and ending at 333.84 °C, and the highest weight loss rate was at 251.28 °C. These results proved that the presence of PULL (saponified) could delay the degradation of PVAc (saponified), which might be due to the strong intermolecular bonding as the crystallization of PULL occurred after saponification, as proven by XRD. Compared with the saponified PULL/PVAc 90/10 film, the saponified PULL/PVAc 80/20 film showed a difference at the second-stage weight loss. The decomposition temperature improved, and the weight loss decreased. The third weight-loss area of the saponified PULL/PVAc 80/20 film occurred within the range of 400 °C–495 °C, with a high weight of residue of 34%. Therefore, the saponified PVAc/PULL 80/20 film showed higher thermal stability than other studied saponified films, which might be due to the higher content of crystallized PULL present in the saponified PVAc/PULL 80/20 film. In addition, all unsaponified films showed higher thermal stability than the saponified films; among all unsaponified films, the thermal stability of PVAc decreased with the addition of PULL (Figure 7 and Figure 8), which might be due to the intermolecular breaking and partial disintegration of the molecular structure.

### 2.9. Stress–Strain Curves

The stress–strain curves for saponified and unsaponified films are shown in Figure 9. Overall, saponification treatment can increase the strength of pure PVAc film and PVAc/PULL blend film, which can be clearly seen from tensile strength results (Figure 9). Compared with the saponified PVAc film, the results showed that the tensile strength of the saponified PVAc/PULL 90/10 film increased from 15 to 45 MPa. Meanwhile, the strength of the PULL/PVAc 80/20 films also increased after saponification compared with the saponified PVAc film. From Figure 9, the tensile strength of the saponified PVAc/PULL 80/20 film was two times higher than the saponified PVAc film and 1.6 times lower than the saponified PVAc/PULL 90/10 film. It is also seen that the failure strain decreased drastically after the saponification, as saponified films became stiff and elongated less. Moreover, all saponified films showed higher young modulus than unsaponified films. Nevertheless, owing to the crystallization of PULL occurring during saponification, which was indicated by the XRD results, the strength of saponified films increased by varying degrees (Figure 9a–c).

### 2.10. Wound-Healing Properties of Saponified PVAc/PULL Film

The wound-healing execution of the saponified PVAc/PULL film was explored via an in vivo test. The representative images of wounds and wound-healing profiles on the 0th, 6th, 9th, and 10th days are shown in Figure 10. On the 6th, 9th, and 10th days, the wound contraction for saponified PVAc/PULL film (saponification time = 72 h) groups had the lead over the control group, medicine (Fucidin) group, and unsaponified PVAc/PULL film (saponification time = 0 h) group. Wounds for PVAc/PULL film (saponification time = 72 h) groups were seen to have a closure on the 9th day. In addition, after 10 days, the PVAc/PULL film (saponification time = 72 h) group treated wounds completely, whereas the control group showed a lead over the medicine and PVAc/PULL film (saponification time = 0 h) groups, which remained at the same level. PVA-based wound dressing shows improved performance in wound healing in the presence of drugs [34], whereas our saponified PVAc/PULL film wound dressing has been confirmed to improve the wound-healing process compared with drugs in all healing stages. All studied data demonstrated that the prepared saponified PVAc/PULL films advanced the wound-healing rate compared with other groups (control, Fucidin, and unsaponified PVAc film groups). The saponified PVA/PULL films helped heal wounds very rapidly due to their surface properties, i.e., high surface area-to-volume ratio and extra hydrophilicity due to the presence of PULL, which accelerated the absorption of exudates. A rough surface should be defined as necessary in order to take into account the material’s suitability for wound healing since it enhances the cell’s adhesion to the film because of its flexible cell membrane. Furthermore, the roughness of the film surface affects the bacteria’s ability to attach [35].

## 3. Materials and Methods

### 3.1. Materials

VAc was purchased from Sigma-Aldrich (Merck Korea, Seoul, South Korea). After purchasing it, first, it was cleaned using NaHSO_4_ (aqueous) solution and water. Afterward, anhydrous CaCl_2_ was used for drying VAc. Finally, it was distilled under reduced pressure and a nitrogen atmosphere. Pure PULL powder (90%) with molecular weight (*M_w_* = 20,000) was purchased from Hayashibara Biochemical Laboratories Inc. (Okayama, Japan). PVA with *M_n_* = 127,000 g/mol and degree of saponification = 88% was also purchased from Sigma-Aldrich (Merck Korea, Seoul, South Korea). An initiator, 2,2′-azobis(2,4-dimethylvaleronitrile) (ADMVN), was collected from Wako Co., Tokyo, Japan; recrystallization was required twice before using. NaOH, Na_2_SO_4_, and MeOH, required for the saponification reaction, were supplied by Duksan (Seoul, South Korea). Dimethyl sulfoxide (DMSO) (Duksan, Seoul, South Korea) was used as a solvent to produce blend films.

### 3.2. Fabrication of PVAc/PULL Blend Film and Heterogenous Saponified PVA/PULL Film

PVAc powder was produced using suspension polymerization, as per our previous study [27]. The prepared PVAc/PULL was fabricated by a casting/solvent evaporation process. Film solutions, including pure PVAc and PVAc/PULL with different blend ratios of 90/10 and 80/20, were fabricated successfully using solvent DMSO, and polymer concentration was kept at 7 wt.% based on the solution weight. The fabrication of a PVAc/PULL film with a blend ratio of 70/30 was also tried, but the filmmaking was unsuccessful due to an extremely uneven surface. After successful fabrication, the prepared solution (20 g) was cast into a smooth and level bottom Teflon Petri dish (100 mm diameter × 10 mm height). Subsequently, the Petri dish containing the solution was placed in a vacuum oven. Initially, the drying temperature was set to 35 °C and gradually increased to 65 °C until drying was complete, which facilitated a smooth surface. In addition, the vacuum was set at a low value (20 mmHg) to produce a bubble-free surface. After complete drying, the films were peeled from the Petri dish and kept in a plastic bag for subsequent experiments. The prepared PVAc/PULL blend film was then converted to PVA/PULL film using the heterogenous saponification process. The heterogenous saponification reaction was performed following a reference with slight modification [16]. For the heterogeneous saponification of the PVAc/PULL film to the PVA/PULL film, an alkali solution (10 g each of NaOH, Na_2_SO_4_, and MeOH, plus 100 g of H_2_O) was prepared and transferred to a 50 mL vial containing a magnetic bar. Rectangular pieces of each film (sizes varied depending on characterization method) were wrapped with the mesh cloth by sewing to prevent floating and twisting in the solution, then inserted into the vial containing alkali solution. The vial was sealed with a cap and placed in a glass water bath set to 50 °C with a magnetic stirring device. The saponification was continued and stopped after 72 h, resulting in transforming from PVAc/PULL film into PVA/PULL film. After the saponification reaction, it was removed from the vial and immersed in cold water, then allowed to rest for 1 min to complete precipitation. The saponified film was then thoroughly rinsed with water and dried at room temperature until completely dry. The saponification time and temperature were set to 72 h and 50 °C. Table 1 presents the thickness, weight, and contact angle of saponified and unsaponified films. A photograph of saponified and unsaponified films is presented in Figure 11.

### 3.3. Mechanical Properties

Film thickness calculations were performed at 10 varying points on every specimen with a thickness gauge (Digital Vernier caliper, Würth Korea, Dongcheon-ro, South Korea). Afterward, a mechanical test was performed according to the following ASTM D638-96 type Π requirements using an Instron 5567 Universal Testing Machine with a crosshead speed of 20 mm/min. The load and displacement were set to approximately 500 N and 200 mm, respectively. The sample size was 2 × 8 cm^2^ (width × length); to obtain the results, the mean value of three specimens was calculated. The tensile strength and elongation percentage were calculated following reference [36].

### 3.4. Characterization

The surface topography of the film surface was examined by optical microscopy (OM), and the surface roughness was measured using atomic force microscopy (AFM). The hydrophilic characteristics of the film were measured using a water contact angle meter (Dino-Lite, AM703MZT, Korea NARAE PLUS Co., Ltd., Seoul, South Korea) according to reference [21]. The PVAc/PULL samples were dissolved in d_6_-dimethyl sulfoxide for ^1^H-NMR analysis (AVANCE III 500, Bruker, Germany). The analysis was performed in the solution state, and the degree of saponification was calculated according to reference [16]. To confirm the conversion of PVAc/PULL to PVA/PULL, their FT-IR spectra were traced in attenuated total reflection mode applying an FT-IR spectrometer (Frontier, Perkin Elmer, Waltham, MA, USA). The scan range was 400–4000 cm^−1^, with a resolution of 4 cm^−1^. Applying an X-ray diffractometer (D/Max–2500, Rigaku, Tokyo, Japan), the XRD patterns of films were also examined to ensure the formation of saponified PVA/PULL film. The voltage applied during scanning was 40 kV, and the scanning speed was 4°/min. To reveal and compare thermal properties between the saponified and unsaponified films, thermogravimetric analysis (TGA) (model Q-50 from TA Instruments, New Castle, DE, USA) was performed. This analysis was accomplished under a nitrogen atmosphere (a heating rate of 10 °C/min in the temperature range of 20–600 °C).

### 3.5. Assessment of In Vivo Wound-Healing Efficiency

The present study was carried out according to reference [34]. First, Sprague-Dawley rats were anesthetized using 0.2 mL/10 g of avertin via i.p. injection, and the dorsal hair of each rat was shaved using a sterilized scissor. Using 70% ethanol, their skins were cleaned and scraped with coarse sandpaper; scratch wounds (8 mm) were formed on the back of each rat with a biopsy punch (8 mm, SPICA, Wooyang Medical Co., Ltd., Seongnam-si, Korea) to introduce bleeding and oozing fluid, which represented injury to only the surface portion of the skin. Every wound was wrapped with gauze (control), the commercial product FUCIDIN, madecassol, and PVAc film (0 and 72 h). All materials were fastened with a flexible adhesive bandage (3M™ Durapore™ Surgical Tape). All rats were individually kept in separate cages. The wounds were observed using a digital camera at preset intervals.

## 4. Conclusions

Fully saponified PVA/PULL (100/0, 90/10, and 80/20) films were obtained from PVAc/PULL (100/0, 90/10, and 80/20) films, respectively, after 72 h of saponification at 50 °C. From this study, we conclude that saponification treatment can enhance the surface properties of films, which is a good indicator of various useful medical applications. In addition, a contact angle study revealed that saponified films became more hydrophilic, and the presence of PULL could yield extra hydrophilicity. Further, the mechanical resistance of saponified films was improved, particularly for blend films, and the maximum (42.41 MPa) occurred for the saponified PVAc/PULL (90/10) film. Failure strain results also proved that the saponified films have a harder and less deformable nature. Moreover, TGA results revealed that among all saponified films, the highest thermal stability (residue 34%) was obtained for the saponified film with the highest PULL content (PVAc/PULL 80/20). In this study, we demonstrate a new approach for fabricating saponified PVA/PULL blend films with improved thermal properties by preparing PVAc/PULL blend films with different ratios, followed by heterogenous saponification. Further, in vivo experiments demonstrated that saponified PVA/PULL film displayed the best-accelerated wound-healing capabilities through the development of re-epithelialization. The basic mechanism of the accelerated wound-healing ability of the prepared wound dressing could be due to improved surface area and hydrophilicity, which could absorb excess wound exudates rapidly as well as prevent outside pathogenic bacteria from entering the wound area and retaining aseptic wound healing surroundings. The alkaline saponified PVA/PULL films displayed significant prospects as a novel wound-healing application.

## Figures and Tables

**Figure 1 ijms-25-01026-f001:**
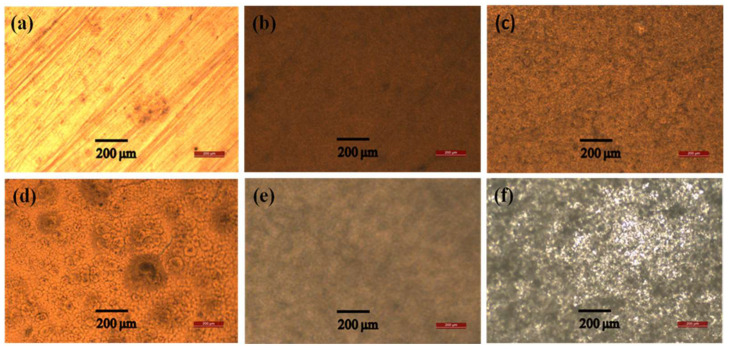
OM images of (**a**–**c**) unsaponified PVAc, PVAc/PULL 90/10, and PVAc/PULL 80/20 films and saponified (**d**–**f**) PVAc, PVAc/PULL 90/10, and PVAc/PULL 80/20 films.

**Figure 2 ijms-25-01026-f002:**
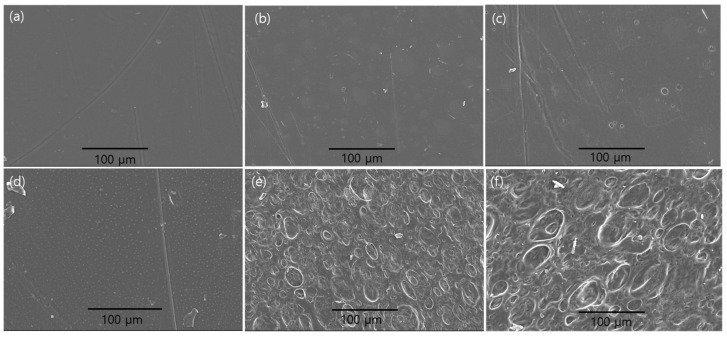
SEM images of (**a**–**c**) unsaponified PVAc, PVAc/PULL 90/10, and PVAc/PULL 80/20 films and saponified (**d**–**f**) PVAc, PVAc/PULL 90/10, and PVAc/PULL 80/20 films.

**Figure 3 ijms-25-01026-f003:**
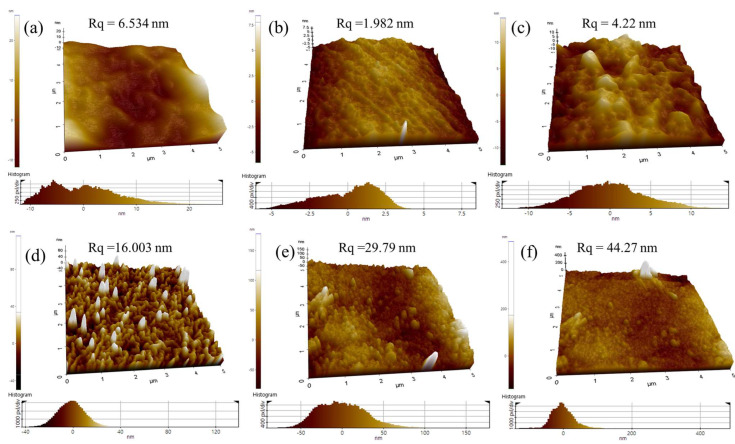
AFM images of (**a**–**c**) unsaponified PVAc, PVAc/PULL 90/10, and PVAc/PULL 80/20 films and saponified (**d**−**f**) PVAc, PVAc/PULL 90/10, and PVAc/PULL 80/20 films.

**Figure 4 ijms-25-01026-f004:**
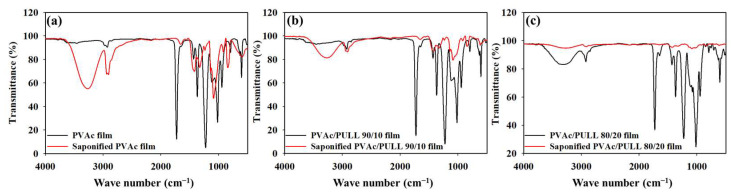
FT-IR data for (**a**) unsaponified and saponified PVAc films, (**b**) unsaponified and saponified 90/10 PVAc/PULL blend films, and (**c**) unsaponified and saponified 80/20 PVAc/PULL blend films.

**Figure 5 ijms-25-01026-f005:**
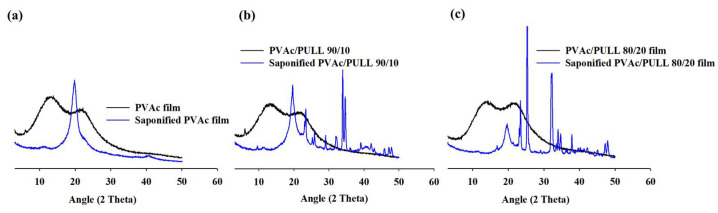
XRD data for (**a**) unsaponified and saponified PVAc films, (**b**) unsaponified and saponified 90/10 PVAc/PULL blend films, and (**c**) unsaponified and saponified 80/20 PVAc/PULL blend films.

**Figure 6 ijms-25-01026-f006:**
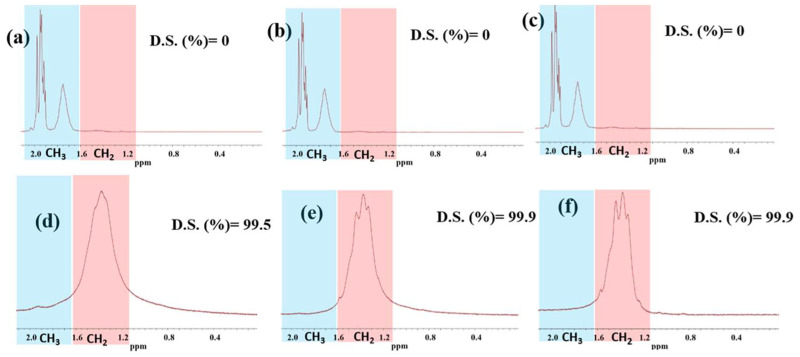
^1^H-NMR data for (**a**–**c**) unsaponified PVAc, PVAc/PULL 90/10, and PVAc/PULL 80/20 films and saponified (**d**–**f**) PVAc, PVAc/PULL 90/10, and PVAc/PULL 80/20 films.

**Figure 7 ijms-25-01026-f007:**
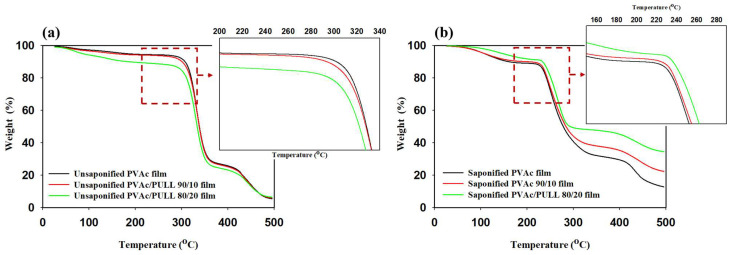
TGA data for (**a**) unsaponified films and (**b**) saponified films.

**Figure 8 ijms-25-01026-f008:**
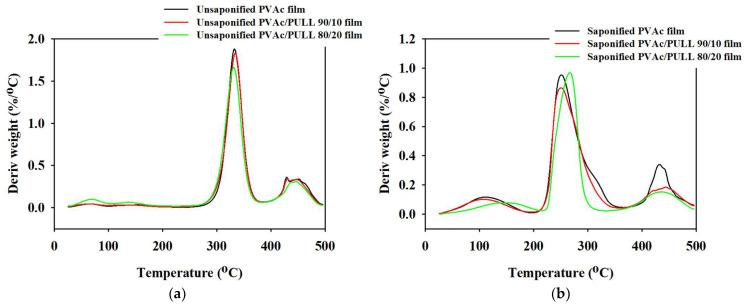
DTG data for (**a**) unsaponified films and (**b**) saponified films.

**Figure 9 ijms-25-01026-f009:**
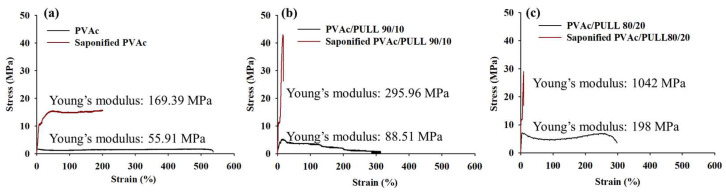
Stress–strain curve for (**a**) unsaponified and saponified PVAc films, (**b**) unsaponified and saponified 90/10 PVAc/PULL blend films, and (**c**) unsaponified and saponified 80/20 PVAc/PULL blend films.

**Figure 10 ijms-25-01026-f010:**
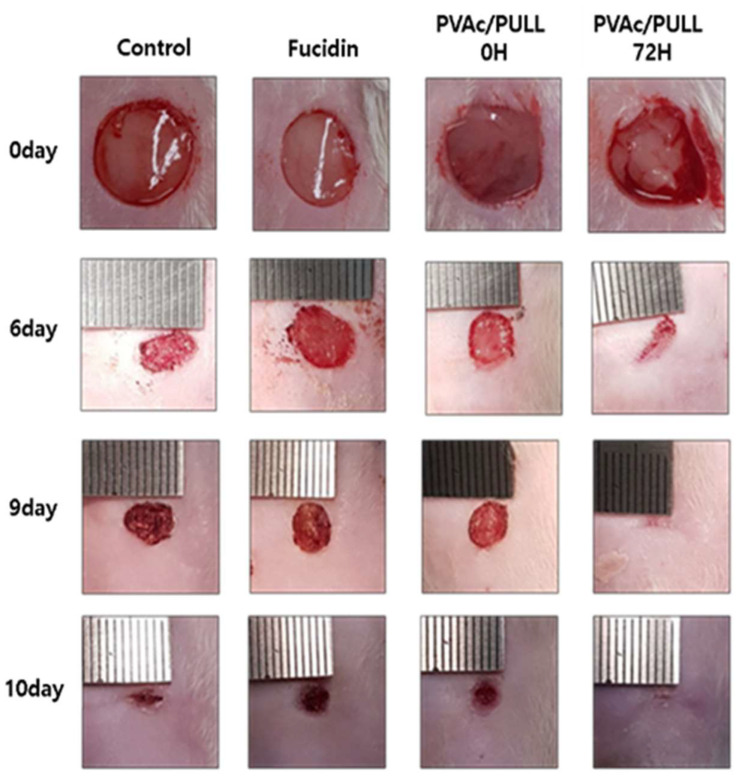
Photographs of wounds on the 0th, 6th, 9th, and 10th days after dressing with saponified PVA/PULL film.

**Figure 11 ijms-25-01026-f011:**
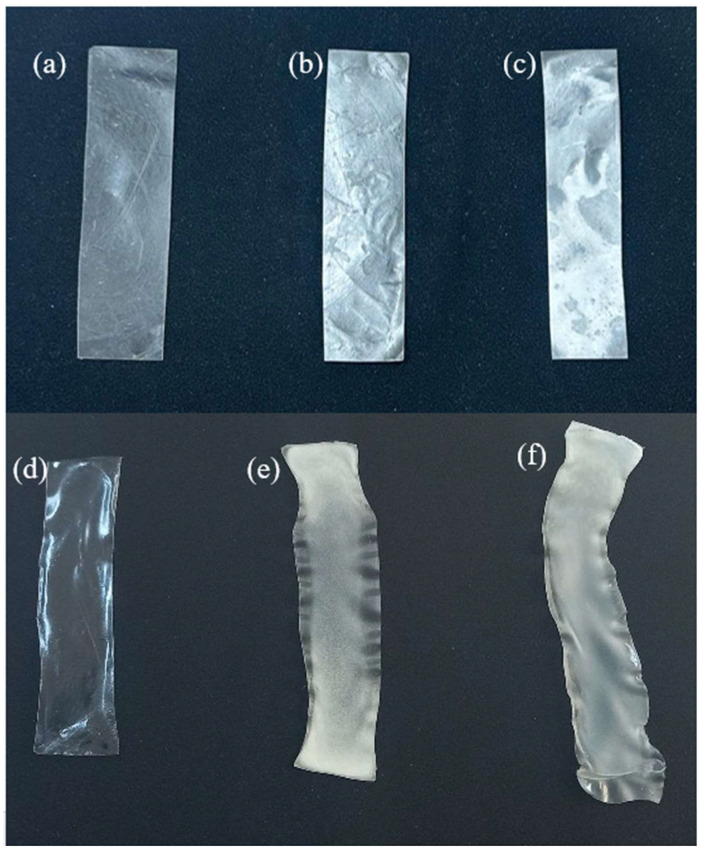
Photographs of (**a**–**c**) unsaponified PVAc, PVAc/PULL 90/10, and PVAc/PULL 80/20 films and saponified (**d**–**f**) PVAc, PVAc/PULL 90/10, and PVAc/PULL 80/20 films.

**Table 1 ijms-25-01026-t001:** Weight, thickness, and contact angle of the saponified and unsaponified films.

Samples	Weight (mg)	Thickness (µm)	Contact Angle (°)
PVAc film	88 ± 7	80 ± 9	85.96 ± 1.76
Saponified PVAc film	39 ± 9	90 ± 6	13.13 ± 0.65
PVAc/PULL 90/10 film	90 ± 5	130 ± 15	81.67 ± 1.98
Saponified PVAc/PULL 90/10 film	62.3 ± 6	90 ± 9	6.73 ± 0. 66
PVAc/PULL 80/20 film	97 ± 16	140 ± 18	62.79 ± 1.44
Saponified PVAc/PULL 80/20 film	58.4 ± 8	97 ±11	4.4 ± 0.44

## Data Availability

Data are contained within the article.

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
