# Peer review of "A Novel Fabrication of Heterogeneous Saponified Poly(Vinyl Alcohol)/Pullulan Blend Film for Improved Wound Healing Application"

_ijms, 2024, doi:10.3390/ijms25021026_

Round 1
Reviewer 1 Report
Comments and Suggestions for Authors
The authors reported the preparation of PVA/pullulan blend film for wound healing application and tested it in in vivo model, which might be interested to the reader. The preparation of PVA/pullulan films using other methods, rather than heterogeneous saponification, have been reported in the literature. However, there is no comparison between the presenting method with other reported methods. This manuscript suffers from minimal discussion. Detail comments are listed below.
1. The title is too long with unnecessary words. Please consider to revise it.
2. Perhaps, providing discussion based on the results is better than using strong words, such as, “remarkable”, “extraordinary”, “exceptional”, etc. These words are used extensively throughout the manuscript.
3. Introduce the mechanism of would healing property of PULL in the introduction should benefit the reader under more about the novelty of this study.
4. Line 42-43, the authors described PULL with “exceptional mechanical behavior” while in line 45, PULL was mentioned with “low mechanical property”. Please revise it.
5. Please provide actual images of the prepared films. SEM images should also be added for evaluation of the surface morphology of the obtained film.
6. Please provide the degree of saponification equation.
7. In the materials and methods section, the author did not provide detail experimental procedure of the heterogenous saponification process. They cited their previous publication on the pure PVA film preparation only. This information is crucial to understand the content of this report.
8. PVA/PULL film has been reported by other group using a different method (Plants 2023, 12(4), 898; International Journal of Biological Macromolecules, 2022, 221, 416–425; Al-Mustansiriyah Journal of Science 2017, 28, 2). What is the advantage of the presenting method?
9. What is the benefit of a thermal-enhanced property as a wound healing assisting film?
10. In the in vivo study, the authors study the different of saponification time on the wound healing property. However, there is no mentioned or results regarding the effect of saponification time on the surface morphology, degree of saponification or mechanical properties.
Comments on the Quality of English LanguageMinor editing of English language required
Author Response
Reviewer 1
The authors reported the preparation of PVA/pullulan blend film for wound healing application and tested it in in vivo model, which might be interested to the reader. The preparation of PVA/pullulan films using other methods, rather than heterogeneous saponification, have been reported in the literature. However, there is no comparison between the presenting method with other reported methods. This manuscript suffers from minimal discussion. Detail comments are listed below.
- The title is too long with unnecessary words. Please consider to revise it.
Answer: Title was revised.
- Perhaps, providing discussion based on the results is better than using strong words, such as, “remarkable”, “extraordinary”, “exceptional”, etc. These words are used extensively throughout the manuscript.
Answer: After a thorough review, the manuscript was edited to eliminate the aforementioned words.
- Introduce the mechanism of would healing property of PULL in the introduction should benefit the reader under more about the novelty of this study.
Answer: We have now discussed the mechanism of PULL's wound-healing property in the introduction.
- Line 42-43, the authors described PULL with “exceptional mechanical behavior” while in line 45, PULL was mentioned with “low mechanical property”. Please revise it.
Answer: The words “exceptional mechanical behavior “are removed.
- Please provide actual images of the prepared films. SEM images should also be added for evaluation of the surface morphology of the obtained film.
Answer: The prepared film's actual images (Figure11) are now included in the manuscript. SEM images are also added.
- Please provide the degree of saponification equation.
Answer: The degree of saponification equation is included in the manuscript.
- In the materials and methods section, the author did not provide detail experimental procedure of the heterogenous saponification process. They cited their previous publication on the pure PVA film preparation only. This information is crucial to understand the content of this report.
Answer: The detail experimental procedure of the heterogenous saponification process is provided.
- PVA/PULL film has been reported by other group using a different method (Plants 2023, 12(4), 898; International Journal of Biological Macromolecules, 2022, 221, 416–425; Al-Mustansiriyah Journal of Science 2017, 28, 2). What is the advantage of the presenting method?
Answer: The prepared heterogeneous saponified PVA/PULL film showed some unique characteristics, such as an uneven surface with high surface area and improved hydrophilic properties, which support the absorption of excess wound exudates and retention of moisture at wound sites.
- What is the benefit of a thermal-enhanced property as a wound healing assisting film?
Answer: The words thermal-enhanced is removed.
- In the in vivo study, the authors study the different of saponification time on the wound healing property. However, there is no mentioned or results regarding the effect of saponification time on the surface morphology, degree of saponification or mechanical properties.
Answer: In vivo test is rectified now to ensure the test results align with the characterized properties of the blend films.
Reviewer 2 Report
Comments and Suggestions for Authors
This manuscript presents the fabrication of a new PVA/pullulan blend film via the heterogeneous saponification process of PVAc/pullulan blend film. The study employs various characterization methods to confirm the successful preparation of these saponified blend films and assess their properties. Through in vivo wound healing tests, the effectiveness of these blend films in wound treatment is demonstrated, showing promise for practical applications. However, the manuscript requires major revisions before publication in the International Journal of Molecular Sciences. The issues requiring careful consideration by the authors are outlined as follows:
1. The authors must ensure strict adherence to journal guidelines for formatting throughout the manuscript, encompassing decimal formatting, capitalization, subscript/superscript usage, and consistent notation (e.g., Fig. or Figs., 72h or 72 h, 50°C or 50 °C).
2. The authors should emphasize the advancements in preparing blend films for wound healing and elucidate the advantages and disadvantages in the introduction section. The authors must also articulate how this work's blend film addresses these concerns.
3. The authors mentioned a more surface roughness of the saponified films, confirmed by AFM and SEM, resulting in their improved wound healing performances. A detailed discussion on the mechanism causing increased surface roughness post-saponification and its impact on wound healing is required.
4. The appearance of additional crystalline peaks and their variations with different blend ratios require thorough explanation and discussion, particularly regarding pullulan crystallization.
5. In section 2.5, discrepancies in the reported saponification degree (DS) require correction and clarification (98% in the text, whereas 99.5% in Figure 5(d)), specifically addressing variations observed in the 1H-NMR spectra.
6. PVA without crosslinking is of moisture-sensitive material. The manuscript should address the moisture stability of the hydrophilic PVAc/pullulan blend films for their practicality in wound healing applications.
7. Surface roughness increasing surface area was mentioned for improved wound healing performances. From this point of view, the extreme unevenness observed in PVAc/PULL 70/30 warrants consideration concerning its impact on wound healing performance, and it should also be considered in this work.
8. In the experimental and characterization parts, the authors focused only on the saponification condition of 72 h 50 °C. However, in the in vivo wound healing tests, the various saponification times of 24-120 h were considered. The discrepancy between saponification times in characterization and in vivo tests needs rectification to ensure the test results align with the characterized properties of the blend films.
Overall Recommendation: Major revision

The manuscript requires minor editing for English language usage. The authors should meticulously review the entire manuscript for language improvements.
Author Response
Reviewer 2
This manuscript presents the fabrication of a new PVA/pullulan blend film via the heterogeneous saponification process of PVAc/pullulan blend film. The study employs various characterization methods to confirm the successful preparation of these saponified blend films and assess their properties. Through in vivo wound healing tests, the effectiveness of these blend films in wound treatment is demonstrated, showing promise for practical applications. However, the manuscript requires major revisions before publication in the International Journal of Molecular Sciences. The issues requiring careful consideration by the authors are outlined as follows:
- The authors must ensure strict adherence to journal guidelines for formatting throughout the manuscript, encompassing decimal formatting, capitalization, subscript/superscript usage, and consistent notation (e.g., Fig. or Figs., 72h or 72 h, 50°C or 50 °C).
Answer: Formatting throughout the manuscript was checked and corrected.
- The authors should emphasize the advancements in preparing blend films for wound healing and elucidate the advantages and disadvantages in the introduction section. The authors must also articulate how this work's blend film addresses these concerns.
Answer: The introduction section now includes a benefit of blend film for wound healing.
- The authors mentioned a more surface roughness of the saponified films, confirmed by AFM and SEM, resulting in their improved wound healing performances. A detailed discussion on the mechanism causing increased surface roughness post-saponification and its impact on wound healing is required.
Answer: Mechanism of increased surface roughness of post–saponification film is already discussed in the section 2.1.
“The micrographs of saponified films displayed higher surface roughness than those of the unsaponified films, which might be due to the dissolution of film skin in the alkaline solution and increase in film density during the drying of the film after saponification’’
Section 2.10 now includes a discussion of how surface roughness affects wound healing.
- The appearance of additional crystalline peaks and their variations with different blend ratios require thorough explanation and discussion, particularly regarding pullulan crystallization.
Answer: A discussion about additional crystalline peak and peak variation is included in the manuscript.
- In section 2.5, discrepancies in the reported saponification degree (DS) require correction and clarification (98% in the text, whereas 99.5% in Figure 5(d)), specifically addressing variations observed in the 1H-NMR spectra.
Answer: In section 2.5, the degree of saponification (DS) is corrected.
- PVA without crosslinking is of moisture-sensitive material. The manuscript should address the moisture stability of the hydrophilic PVAc/pullulan blend films for their practicality in wound healing applications.
Answer: Moisture stability of the heterogeneous saponified hydrophilic PVAc/pullulan blend films was discussed with a reference in the introduction section.
- Surface roughness increasing surface area was mentioned for improved wound healing performances. From this point of view, the extreme unevenness observed in PVAc/PULL 70/30 warrants consideration concerning its impact on wound healing performance, and it should also be considered in this work.
Answer: The fabrication of a PVAc/PULL film with a blend ratio of 70/30 was also tried, but the film making was unsuccessful due to an extremely uneven surface. That’s why 70/30 ratio was not studied in this study.
- In the experimental and characterization parts, the authors focused only on the saponification condition of 72 h 50 °C. However, in the in vivowound healing tests, the various saponification times of 24-120 h were considered. The discrepancy between saponification times in characterization and in vivo tests needs rectification to ensure the test results align with the characterized properties of the blend films.
Answer: To make sure the test findings match the blend films' described properties; the in vivo tests have been corrected.
Round 2
Reviewer 1 Report
Comments and Suggestions for Authors
The authors have diligently revised the manuscript to address the reviewers' comments and suggestions. However, there remain a few minor enhancements needed to refine the manuscript further:
1. References should be included for Line 42-44 and Line 54-61.
2. Please consider enlarging the text size in Figures 3, 4, 5, 7, and 9 for better readability.
3. It would be beneficial to include a discussion highlighting the advantages of the proposed reporting method in comparison to other methodologies within the manuscript.
Comments on the Quality of English LanguageMinor editing of English language required
Author Response
Dear Editor,
We appreciate you and the reviewers for your precious time in reviewing our paper and providing valuable comments. It was your valuable and insightful comments that led to possible improvements in the current version. The authors have carefully considered the comments and tried our best to address every one of them. We hope the manuscript after careful revisions meet your high standards. The authors welcome further constructive comments if any.
Revisions to the manuscript are marked up using the “Track Changes” function.
Sincerely
Reviewer 1
The authors have diligently revised the manuscript to address the reviewers’ comments and suggestions. However, there remain a few minor enhancements needed to refine the manuscript further.
- References should be included for line 42-44 and line 54-61.
Answer: Reference is included for line 42-44 and line 54-61.
- Please consider enlarging the text size in Figures 3,4,5,7 and 9 for better readability
Answer: Text size is enlarged in Figure 3,4,5,7 and 9.
- It would be beneficial to include a discussion highlighting the advantage of the proposed reporting method in comparison to other methodologies within the manuscript.
Answer: A discussion about the advantages of the saponification method in comparison to other methods is included in the introduction part.
Additional:
- New references were added.
- The relevancy of reference was checked and corrected where necessary.
- English was reviewed and adjusted as needed.
Reviewer 2 Report
Comments and Suggestions for Authors
The manuscript was well-revised.
Author Response
Reviewer 2
- The manuscript was well-revised.
Answer: English was reviewed and adjusted as needed.